# A New Regioselective Synthesis of the Cysteine-Rich Peptide Linaclotide

**DOI:** 10.3390/molecules28031007

**Published:** 2023-01-19

**Authors:** Zhonghao Qiu, Xiandong Dai, Chongxu Fan, Ying Cao, Zirui Lv, Xingyong Liang, Fanhua Meng

**Affiliations:** 1State Key Laboratory of NBC Protection for Civilian, Beijing 102205, China; 2School of Chemistry Engineering, Sichuan University of Science & Engineering, Zigong 643000, China

**Keywords:** linaclotide, regioselective synthesis, disulfide bridge

## Abstract

Linaclotide is a 14-amino acid residue peptide approved by the FDA for the treatment of irritable bowel syndrome with constipation (IBS-C), which activates guanylate cyclase C to accelerate intestinal transit. Here we show a new method for the synthesis of linaclotide through the completely selective formation of three disulfide bonds in satisfactory overall yields via mild oxidation reactions of the solid phase and liquid phase, using 4-methoxytrityl (Mmt), diphenylmethyl (Dpm) and 2-nitrobenzyl (O-NBn) protecting groups of cysteine as substrate, respectively.

## 1. Introduction

Guanylate cyclase C (GC-C) is an important receptor protein expressed by intestinal epithelial cells, and its dysregulation leads to severe intestinal diseases. Linaclotide is a GC-C receptor agonist and was approved by the US FDA in August 2012 for the treatment of adult chronic idiopathic constipation and irritable bowel syndrome with constipation (IBS-C), which not only accelerates intestinal transit but also improves abdominal pain [1,2,3,4]. Linaclotide is a 14-amino acid residue peptide that contains three disulfide bonds lying between Cys^1^-Cys^6^, Cys^2^-Cys^10^, and Cys^5^-Cys^13^, structurally analogous to the diarrhea-causing, heat-stable enterotoxins produced by *E. coli.* [5,6] (Figure 1).

Disulfide bonds are crucial for maintaining the biological activity and structure of peptides and proteins [7,8]. The misfolding of proteins has been associated with numerous diseases, one type of misfolding being the mispairing of disulfide bridges [9,10,11,12,13]. The challenge in the assembly of linaclotide consists of achieving the correct and clean folding of its three disulfide bridges. At present, the formation of disulfide bonds in linaclotide is based on the random oxidation, semi-regioselective oxidation and regioselective methods. In the random oxidation [14,15,16,17,18], the synthetic steps are simple, and only one kind of protecting group for Cys is used. However, for peptides where three disulfide bonds need to be formed site-specifically, many different disulfide bond mismatched isomers will be obtained. Although it is possible to make the conversion into target molecules in the oxidation process as high as possible by some buffer systems, the disulfide bond mismatched isomer impurities still cannot be avoided to produce and the yield is lower. In the semi-selective oxidative synthesis of linaclotide [19,20,21,22], one disulfide bond is site-specifically oxidized which decreases the number of different isomers formed, as compared with the random oxidation method. However, the production of mismatched isomers still cannot be avoided completely. The regioselective oxidative method is based on the stepwise formation of disulfide bonds employing orthogonally protected cysteine (Cys). In general, disulfide-rich peptides synthesized by the regioselective oxidative method can obtain the correct pairing of Cys. However, only a few papers and patents reported the regioselective oxidative method to synthesize linaclotide [23,24,25]. Furthermore, the disadvantage of the regioselective oxidative method is that the final compound is obtained with unsatisfactory yields, because of premature loss of peptide during multiple purification steps. Brik reported a one-pot method to synthesize linaclotide [25]. They relied on small molecule activation of the Cys side chain via the disulfiram (DSF) and ultraviolet (UV) light/Pd chemoselective chemistries for one-pot and ultrafast formation of three disulfide bonds (Figure 2). Nevertheless, Pd is a heavy metal as a reaction material, which is not conducive to peptide drug synthesis. We herein present the first and facile synthesis of linaclotide via the regioselective oxidative synthetic approach, which is an effective and environmental-friendly synthetic strategy.

## 2. Results and Discussion

We set out to design an effective set of sequential reactions guided by the following consideration. First, three disulfide bonds of linaclotide were formed by completely selective oxidation, thereby enabling the efficient site-specific synthesis of three completely intercrossed disulfide bonds. Second, the oxidant should be moderated to eliminate the formation of side products. Third, fewer purification steps must be required to avoid the significant loss of material.

A recent study has shown that 2-nitrobenzyl (*o*-NBn) photosensitive PG of Cys can induce a fast (within 8 min) and simultaneous decaging and selective disulfide formation when it was exposed to UV light in presence of DSF at pH 7 [25,26]. We tried to introduce 2-nitrobenzyl (*o*-NBn) photosensitive PG of Cys into the synthesis of linaclotide.

To achieve the above object, we designed the following technical solutions (Figure 3): (1) synthesizing linaclotide precursor resin through Fmoc-strategy solid phase synthesis; (2) forming the first disulfide bond through on-resin oxidation; (3) forming the second disulfide bond through liquid phase oxidation after cleavage; and (4) deprotecting 2-nitrobenzyl of cysteine side chain by UV light, and oxidatively coupling the third disulfide bond to obtain linaclotide.

First of all, the Cys modified with *o*-NBn was prepared from the L-cysteine by the two-step nucleophilic substitution reaction via activation with trimethylchlorosilane (Figure 4, see Appendix A of the experimental details).

We prepared the linear linaclotide peptide by solid phase peptide synthesis (SPPS), wherein the side chains of Cys corresponding to positions 1 and 6 of linaclotide were protected by 4- methoxytrityl (Mmt), the side chains of Cys corresponding to positions 2 and 10 were protected by diphenylmethyl (Dpm), and the side chains of Cys corresponding to positions 5 and 13 were protected by 2-nitrobenzyl (o-NBn). Remove Mmt protecting groups from the peptide resin with 2% mixed solution of TFA/DCM as a deprotecting agent, and the reaction endpoint was a change in the solution from red to colourless. Then two free thiols were oxidized via *N*-chlorosuccinimide (NCS) as an oxidizing agent to form the first disulfide bond, a linaclotide precursor resin containing a mono-dithio ring was obtained. The mono-dithio cyclic peptide was cleaved from the resin and Dpm protecting groups were simultaneously removed by lysing mixture of TFA, H_2_O, TRIPS (triisopropylsilane) in a volume ratio of 95: 2.5: 2.5 (Figure 3a). Purification was carried out by reverse phase high performance liquid chromatography. We designed the following synthetic one-pot operation for the formation of the other two disulfide bonds. The second disulfide bond was formed through oxidizing the mono-dithio cyclic peptide with disulfiram (DSF) as oxidizing agent, and the bis-dithio cyclic peptide was obtained (Figure 3b). Then the *o*-NBn protecting groups of Cys were removed from the bis-dithio cyclic peptide when it was exposed to 365 nm UV light in presence of DSF at pH 7, and simultaneously oxidizing to form the third disulfide bond to generate the linaclotide (Figure 3c).

Through the monitoring of the reaction process using HPLC, we observed removing protective groups and forming disulfide bonds via photo reaction as the key steps, which affected the overall synthesis yield. Thus, we tried to optimize the reaction conditions. We used to form a pair of disulfide bonds for synthesis of the polypeptide drug vasopressin via ultraviolet light reaction from Cys (*o*-NBn) precursor (see Appendix A). The result of the experiments showed that the optimal reaction condition was DSF as the oxidation reagent, 50% ACN/H_2_O (acetonitrile/water) as the solvent and exposed to 365 nm UV light. In order to improve ultraviolet light reaction efficiency, and further optimize the reaction conditions of formation the disulfide bond of linaclotide by ultraviolet light reaction basis on the previous research, we investigated the effects of light area, light intensity (controlled by current) and light duration on the yield. Although the raw materials (bis-dithio cyclic peptide) gradually disappear with the increase in light reaction time when the current intensity was 5A and light area was 0.785 cm^2^ of 5 mL test tube, no reaction products was detected (Table 1, entries 1–5). When we increased the current intensity to 8A, there were a few products generation, and the raw materials and product gradually disappear with the increase in light reaction time (Table 1, entries 6–10). It is obvious that using increaser light intensity is beneficial to improve the yield of the linaclotide. When we increased the current intensity to 10 A, the yield of the linaclotide was improved with the longer light duration (Table 1, entries 11–13). It was worth mentioning that the isolate yield of linaclotide was up to 46% after 25 min of UV light (Table 1, entry 13). However, with the further extension of the illumination time, both the raw materials and products were disappeared (Table 1, entry 14). We speculated that a long exposure of strong UV light would decompose the product and the raw material or occur other side reactions. A similar phenomenon was observed when the current intensity continues to increase to 12A (Table 1, entries 16–18). When we increased the light area by 9 times to 7.065 cm^2^ of 10 mL round bottom flask, the isolate yield of linaclotide was 45% (Table 1, entry 15). The experimental results showed that the light area had little effect on the yield of linaclotide. Therefore, the optimized reaction conditions are that DSF as the oxidation reagent, 50% ACN/H_2_O as solvent and exposed to 365 nm UV light under 10 A current intensity for 25 min.

Finally, we introduced a new regioselective synthetic scheme to preparation of linaclotide. Compared with traditional regioselective synthesis methods containing multistep purification [23,24], our method to form the second and the third disulfide bonds via one-pot reaction, reduced purification steps, lowered premature losses, simplified operation process and improved the yield. Moreover, deprotection and formation of the third disulfide bond under the UV light in one step simultaneously, and no other reagents added, which is easy to handle, less costly, and more economical and environmental-friendly. It also had some advantages in comparison with Brik’s method [25], such as less by-products in the one-pot reaction due to using the pure linacoltide precursor with one disulfide bond as reactant, avoiding the use of heavy metal pd.

## 3. Materials and Methods

### 3.1. Materials and Instruments

Unless otherwise noted, all reagents and solvents were used directly as commercially received. Linear linaclotide peptide synthesis was performed on CEM Liberty automatic microwave peptide synthesizer. HPLC analysis data were recorded on the Agilent 1260 analytical HPLC instrument using Vydac218TP54 C18 (5 μm, 4.6 × 100 mm) column, with 0.1%TFA-water and 0.1%TFA-acetonitrile as mobile phase. Purification was performed by Agilent 1260 preparative HPLC instrument using Agilent Prep (10 μm, 30 × 250 mm) column, and semi-preformed C18 (5 μm, 10 × 250 mm) column, with 0.1%TFA-water and 0.1%TFA-acetonitrile as mobile phase and gradient elution. MS data were determined on an Agilent Ultimate 3000-QE Focus MS mass spectrometer. Lyophilization was carried out on Christ Alpha1–4 Freeze Dryer.

### 3.2. Synthesis of Protected Linear Linaclotide Precursor Resin

The synthesis was carried out using Fmoc-SPPS on a Fmoc-Tyr(*t*Bu)-Wang resin (0.64 mmol/g, 0.25 mmol scale). Peptide synthesis was performed on peptide synthesizer in presence of 4 equiv. of amino acid, HBTU, and 8 equiv. of N, N’- diisopropylethylamine (DIEA). A solution of 20% piperidine in DMF was used to remove the Fmoc-protecting group. After the reaction completed, we obtained the linear linaclotide peptide resin bearing two Cys (1 and 6) modified with Dpm, two Cys (2 and 10) modified with Mmt and the others Cys (5 and 13) modified with 2- nitrobenzyl (*o*-NBn). The resin was washed three times with DMF (10 mL/time). An aliquot of 10 mL of methanol was added to shrink the resin for 30 min, then methanol was suctioned away, and vacuum drying was performed to obtain 610 mg of resin.

### 3.3. Removal of Protecting Group Mmt

An amount of 100 mg of the linaclotide precursor resin obtained in Section 3.2 was swelled in 10 mL of DCM solution for 1 h, the solution was then suctioned away under reduced pressure, and the resin was washed twice with DCM (10 mL/time). The resin was washed with 5 mL of 2% TFA/DCM (*v*/*v*) solution for 2 min each time until the color of the resin changed from red to colorless, then washed twice with DCM (10 mL/time) and washed twice with methanol (10 mL/time). Then, the solution was suctioned away under reduced pressure.

### 3.4. Synthesis of Linaclotide Precursor Resin with Single Disulfide Bond

An aliquot of 5 mL of DMF was added to the linaclotide precursor resin without Mmt protecting group, and then NCS (5. 4 mg, 0.04 mmol) was added. After reacted for 0.5 h, the solution was suctioned away under reduced pressure and the resin was washed three times with DMF (10 mL/time). An aliquot of 10 mL of methanol was added to shrink the resin for 30 min, then methanol was suctioned away, and vacuum drying was performed to obtain 85 mg of resin.

### 3.5. Synthesis of Linaclotide with Single Disulfide Bond

An amount of 85 mg of the resin obtained in 3.4 was added to a 10 mL round bottom flask, 4 mL of a preformulated solution TFA: H_2_O: TRIPS = 95: 2.5: 2.5 (*v*/*v*/*v*) was added and reacted at room temperature for 2 h. The resin was suction filtered and the filtrate was collected. The resin was washed with a small amount of TFA and the filtrates were combined. Then the filtrate was evaporated to 2 mL and the crude peptide was precipitated with precooled Et_2_O (20 mL). The solid was washed three times with precooled Et_2_O (20 mL/time), dissolved in ACN-H_2_O (1:1) (25 mL), and lyophilized. The HPLC analysis was carried out on a C18 analytical column using a gradient of 0–50% B with 1 mL/min, the monitor wavelength was 214 nm. For preparative HPLC, the C18 column in a gradient of 0–50% B with 20 mL/min was used to provide the mono-dithio cyclic linaclotide precursor peptide in 70% yield (18 mg).

### 3.6. Synthesis of Linaclotide

1.7 mg (1 μmol)of mono-dithio cyclic peptides were dissolved in 10 mL of 50% acetonitrile/water (*v*/*v*) solution, 15 equiv. DSF (4.4 mg) was added and reacted at room temperature for 2 h. After the reaction was completed by HPLC monitoring, the mixture was directly exposed to UV irradiation at 365 nm for 25 min (the reaction liquid level was 5 cm away from the light source). Subsequently, the mixture was continued to react for 2 h at room temperature without UV light source. Purification was performed by HPLC using a semi-preparative C18 column (10 × 250 mm) in a gradient of 0–50% B with 2 mL/min. Target peak fractions were collected, concentrated, and lyophilized to give a 46% yield (0.66 mg).

## 4. Conclusions

In summary, we introduced here a combination of solid and liquid phase reaction synthetic schemes to synthesize linaclotide applying activation of the Cys side chain by NCS, DSF, and UV light for the chemoselective and regioselective formation of three disulfide bonds. Furthermore, forming the first disulfide bond through on-resin oxidation gives rise to a pure linaclotide precursor containing one disulfide bond obtained after splitting and purifying. In addition, the deprotection and formation of the third disulfide bond under the UV light in one step simultaneously, which is performed under mild conditions, low in cost, high in overall yield, high in product purity, simple and stable in processes. The synthetic method exhibits a widespread application prospect in the technical field of polypeptide drug synthesis.

## Figures and Tables

**Figure 1 molecules-28-01007-f001:**
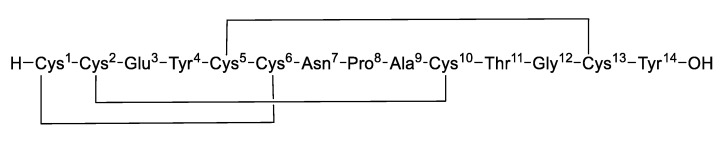
Structure of linaclotide.

**Figure 2 molecules-28-01007-f002:**
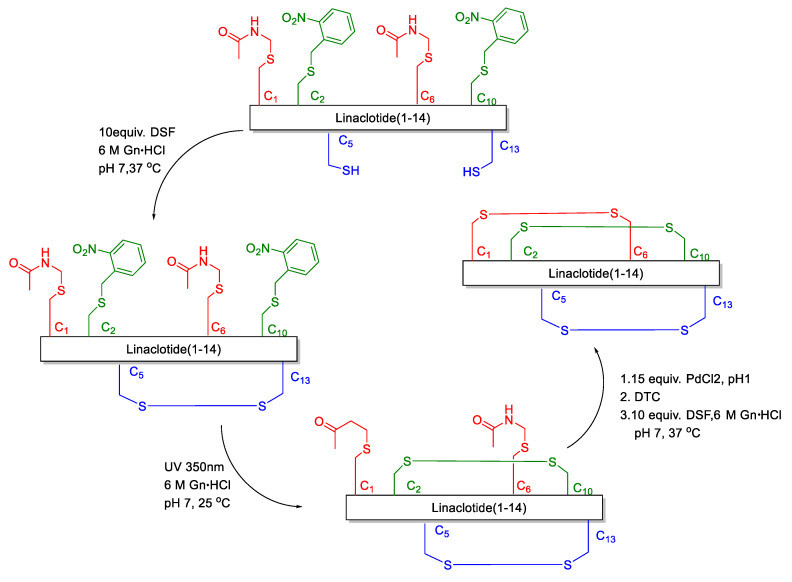
Linaclotide synthesis of Brik’s group.

**Figure 3 molecules-28-01007-f003:**
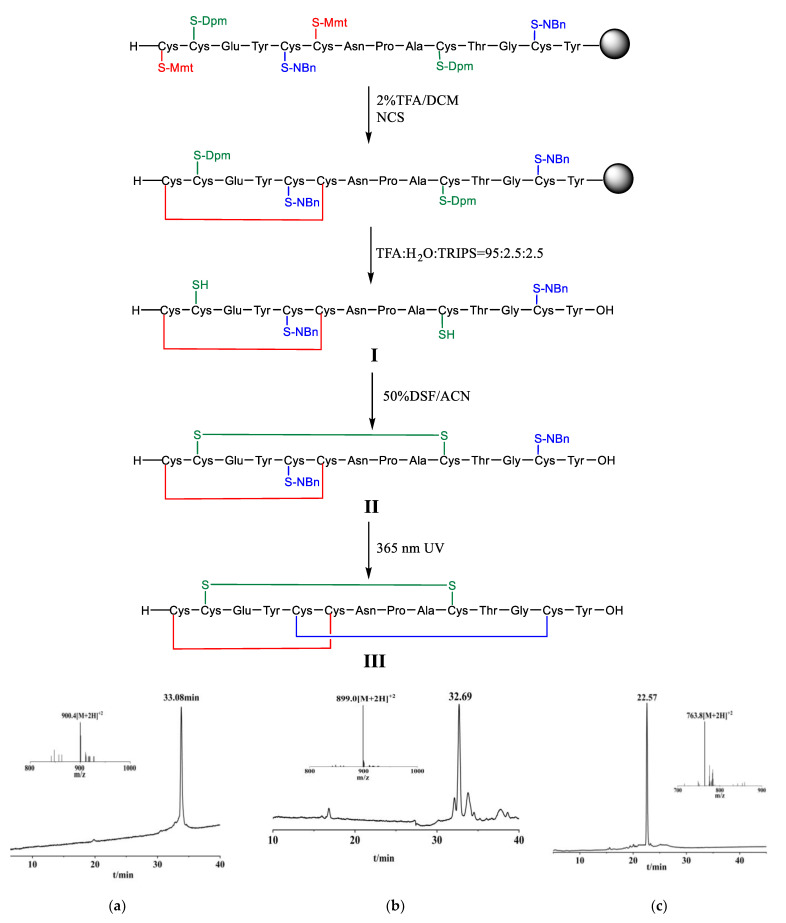
Linaclotide synthesis. (**a**) The main peak corresponds to Linaclotide with a single disulfide bond (I)with the observed mass 900.40 Da, calcd. 900.74 Da [M + 2H]^2+^. (**b**) The main peak corresponds to Linaclotide with two disulfide bonds (II )with the observed mass 899.07 Da [M+2H]^2+^, calcd. 899.73 Da. (**c**) The main peak corresponds to Linaclotide (III) with the three disulfide bonds with the observed mass 763.9 Da [M + 2H]^2+^, calcd. 763.70 Da. Dpm: diphenylmethyl, Mmt: 4-methoxytrityl, NBn: 2-nitrobenzyl, NCS: *N*-chlorosuccinimide, TFA: trifluoroacetic acid, TRIPS: triisopropylsilane, DCM: dichloromethane, DSF: disulfiram, ACN: acetonitrile.

**Figure 4 molecules-28-01007-f004:**
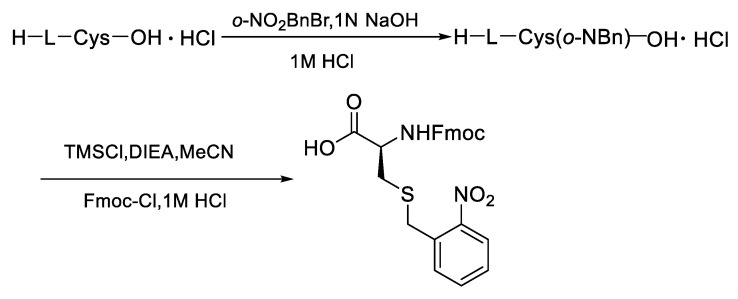
Synthesis of Fmoc- L -Cys(S-o-NBn)-OH.

**Table 1 molecules-28-01007-t001:** Optimizing reaction conditions for formation of the third disulfide bond from UV light reaction.

Entry	Light Area (cm^2^)	Current Intensity (A)	Light Duration (min)	Reaction Result
1	0.785	5	15	A
2	0.785	5	20	A
3	0.785	5	25	A
4	0.785	5	30	B
5	0.785	5	60	B
6	0.785	8	15	A
7	0.785	8	20	B
8	0.785	8	25	C
9	0.785	8	30	C
10	0.785	8	60	D
11	0.785	10	15	E
12	0.785	10	20	F, 30% isolate yield
13	0.785	10	25	F, 46% isolate yield
14	0.785	10	30	D
15	7.065	10	25	F, 45% isolate yield
16	0.785	12	15	E
17	0.785	12	20	F, 35% isolate yield
18	0.785	12	25	D

A: No reaction; B: Raw material reduced without product generation; C: Raw material disappeared and a few products generation; D: Both raw material and product disappeared; E: Raw material reduction and a few products generation; F: Raw material disappeared.

## Data Availability

Not applicable.

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
