# Peer review of "A New Regioselective Synthesis of the Cysteine-Rich Peptide Linaclotide"

_molecules, 2023, doi:10.3390/molecules28031007_

Round 1

Reviewer 1 Report

In manuscript, Liang, meng, and co-workers reported a combination of solid and liquid phase reaction process to synthesize linaclotide via sequential formation of three disulfide bonds. The first one is realized on resin with low concentration TFA deprotection and the NCS oxidation. The second one was established after cleavage from resin and oxidation with DSF in solution and the last one is also in solution after UV light for chemoselective deprotection followed by oxidation. The synthetic strategy features mild conditions, low in cost, high in overall yield, high in product purity, simple and stable in processes. Thus, it can be accepted for publication after a minor revision.

Some corrections are suggested.

14 amino acid peptide  -   14 amino acid residue peptide

-Cys2-cys10   --   Cys2-cys10

Synthesis steps   -   synthetic steps

complete selective oxidation   --  completely selective oxidation

it’s exposed   -   it was exposed

Author Response

In manuscript, Liang, meng, and co-workers reported a combination of solid and liquid phase reaction process to synthesize linaclotide via sequential formation of three disulfide bonds. The first one is realized on resin with low concentration TFA deprotection and the NCS oxidation. The second one was established after cleavage from resin and oxidation with DSF in solution and the last one is also in solution after UV light for chemoselective deprotection followed by oxidation. The synthetic strategy features mild conditions, low in cost, high in overall yield, high in product purity, simple and stable in processes. Thus, it can be accepted for publication after a minor revision. 

Some corrections are suggested.

14 amino acid peptide  -   14 amino acid residue peptide

-Cys2-cys10   --   Cys2-cys10

Synthesis steps   -   synthetic steps

complete selective oxidation   --  completely selective oxidation

it’s exposed   -   it was exposed

Response: Thank you for your careful reading, helpful comments, and constructive suggestions, which has significantly improved the presentation of our manuscript. The errors you pointed out have been corrected.

Reviewer 2 Report

The manuscript by Liang, Meng et al describes a new regioselective synthesis of Linaclotide, a 14-amino acid residue peptide that is used for the treatment of irritable bowel syndrome with constipation (IBS-C). The synthesis features completely selective formation of three disulfide bonds in satisfactory overall yields via mild oxidation reactions of solid phase and liquid phase. I recommend its publication, however, after the authors address the following minor revisions.

1. The abbreviations Dpm, Mmt, NCS, DSF, ACN and others should be explained in the title of Figure 2, not simply in the main text, in order to let the readers easily get to the scientific point of this manuscript. Most readers would like to take a glance at the figures rather than carefully read the text.

2. No definition of ACN was given. What is this compound?

3. In Figure 2, the yield of every single step should be indicated.

4. In Figure 1 and Figure 2, the amino acid residues are indicated by the abbreviations, no functional groups are shown. It is difficult for the readers, especially those not in this field, to understand the reactions. Thus, I suggest the authors include a full structure of the peptide with all the functionality visible. Again Figure 2, the functional groups that participate the reactions should be shown.

5. Experimental section 3.5 and 3.6, how were the yields of the products calculated? Did the authors measure their mass?

Author Response

The manuscript by Liang, Meng et al describes a new regioselective synthesis of Linaclotide, a 14-amino acid residue peptide that is used for the treatment of irritable bowel syndrome with constipation (IBS-C). The synthesis features completely selective formation of three disulfide bonds in satisfactory overall yields via mild oxidation reactions of solid phase and liquid phase. I recommend its publication, however, after the authors address the following minor revisions.

  1. The abbreviations Dpm, Mmt, NCS, DSF, ACN and others should be explained in the title of Figure 2, not simply in the main text, in order to let the readers easily get to the scientific point of this manuscript. Most readers would like to take a glance at the figures rather than carefully read the text.

Response: Thank you for pointing out this problem in manuscript. We have added the full name of these abbreviations in the title of Figure.

  1. No definition of ACN was given. What is this compound?

Response: Thank you for pointing out this problem in manuscript. We have added the full name of ACN in the main text and the title of Figure.

  1. In Figure 2, the yield of every single step should be indicated.

Response: It is difficult to calculate the yield after reaction on solid resin. However, after cracking from the resin, we calculated the corresponding yields through the mass of the purified lyophilized products and the initial reaction amount, and gave them in the experimental section 3.5 and 3.6.

  1. In Figure 1 and Figure 2, the amino acid residues are indicated by the abbreviations, no functional groups are shown. It is difficult for the readers, especially those not in this field, to understand the reactions. Thus, I suggest the authors include a full structure of the peptide with all the functionality visible. Again Figure 2, the functional groups that participate the reactions should be shown.

Response: Thank you for pointing out this problem in manuscript. In order to more intuitive presentation the composition of linaclotide in the Figure 1, we used the English abbreviations of amino acids to express its composition due to its chemical structural formula is more complex. But we have added functional groups of the linaclotide that participate the reactions in the Figure 2, such as -SH. And we used the English abbreviations of amino acids instead of single letter abbreviations, which make the reader who not in this field can understand the reaction as easy as possible.

  1. Experimental section 3.5 and 3.6, how were the yields of the products calculated? Did the authors measure their mass?

Response: The yields of the experimental section 3.5 and 3.6 were obtained by measure the mass of product after lyophilization respectively. And we added the mass of products in the experimental section 3.5 and 3.6.

Reviewer 3 Report

Manuscript Title: A New Regioselective Synthesis of the Cysteine-rich Peptide Linaclotide  

Manuscript Number: molecules-2156774

Article Type: Research Article

Comments:

The manuscript “A New Regioselective Synthesis of the Cysteine-rich Peptide Linaclotide” by Fanhua Meng and co-workers, developed a regio-selective synthesis of Linaclotide that contains multiple cysteines.

The authors well drafted this manuscript to describe efficient synthesis for Linaclotide, which contains 14-amino acid residues, and is clinically approved by the FDA for the treatment of irritable bowel syndrome with constipation (IBS-C). The novel method for the synthesis of linaclotide is very elegant and this approach proceeds through the selective formation of three disulfide bonds in overall good yields. The conditions are very mind and performed solid phase and liquid phase synthesis with selective protecting groups on cysteine.    

Overall, I thoroughly enjoyed reading this draft along with the good balance of supporting data in the results and discussions. Good to see the data for the synthesis of peptide intermediates, final compounds data, experimental, and HPLC data. The results described in this draft are more appropriate and convincing with the experimental data. Furthermore, this work is high relevance and suitable for publication in the ‘Molecules’ journal. I strongly accept this draft for publication with minor revisions. However, I wish to comment and give some suggestions for improvising the quality of the manuscript.

Modest items that need to be address include:

1.       It would be good if could add Brik and other research group’s work as the schematic diagrams in the introduction part.

2.       Can you comment on whether late-stage solution phase synthesis to perform the UV reaction is practical for bulk synthesis?

3.       Figure 2, the Wang resin release the C-terminal acid and not amine. Please correct it.

4.       In Scheme 2, is Dpm protecting group cleaved with 95% TFA parallel with Wang resin?

Author Response

The manuscript “A New Regioselective Synthesis of the Cysteine-rich Peptide Linaclotide” by Fanhua Meng and co-workers, developed a regio-selective synthesis of Linaclotide that contains multiple cysteines.

The authors well drafted this manuscript to describe efficient synthesis for Linaclotide, which contains 14-amino acid residues, and is clinically approved by the FDA for the treatment of irritable bowel syndrome with constipation (IBS-C). The novel method for the synthesis of linaclotide is very elegant and this approach proceeds through the selective formation of three disulfide bonds in overall good yields. The conditions are very mind and performed solid phase and liquid phase synthesis with selective protecting groups on cysteine.    

Overall, I thoroughly enjoyed reading this draft along with the good balance of supporting data in the results and discussions. Good to see the data for the synthesis of peptide intermediates, final compounds data, experimental, and HPLC data. The results described in this draft are more appropriate and convincing with the experimental data. Furthermore, this work is high relevance and suitable for publication in the ‘Molecules’ journal. I strongly accept this draft for publication with minor revisions. However, I wish to comment and give some suggestions for improvising the quality of the manuscript.

Modest items that need to be address include:

  1. It would be good if could add Brik and other research group’s work as the schematic diagrams in the introduction part.

Response: Thank you for your careful reading, helpful comments, and constructive suggestions, which has significantly improved the presentation of our manuscript. We have added the Brik group’s work by schematic diagram in the introduction part.

  1. Can you comment on whether late-stage solution phase synthesis to perform the UV reaction is practical for bulk synthesis?

Response:  At present, there are some industrial UV photocatalytic chemical reactors, as shown in the figure below(https://www.ekato.com.cn/products/industrial-photoreactors/), so the late-stage solution phase synthesis by the UV reaction may be expected to achieve in industrial production.

  1. Figure 2, the Wang resin release the C-terminal acid and not amine. Please correct it.

Response: Thank you for pointing out this error in manuscript and it has been corrected.

  1. In Scheme 2, is Dpm protecting group cleaved with 95% TFA parallel with Wang resin?

 Response: Yes, Dpm protecting group can be cleaved with 95% TFA parallel with Wang resin.